# GSLLM: GEOSPATIAL KNOWLEDGE ACQUISITION FOR LARGE LANGUAGE MODEL

## ABSTRACT

Geospatial information and its associated inferences play a critical role in numerous real-world applications. Although large language models (LLMs) acquire extensive general knowledge through large-scale pretraining, they typically lack explicit representations of geospatial data. In this study, we propose a novel framework for enabling LLMs to acquire and utilize geospatial knowledge. By introducing a set of specialized tokens designed to represent geospatial entities—such as coordinates, locations, and addresses—we effectively embed geospatial information into the model's token space. Building upon this enhanced representation, we conduct supervised fine-tuning (SFT) and reinforcement learning (RL) on a pretrained geospatially augmented model to evaluate its performance across multiple downstream tasks. Our approach demonstrates a systematic method for integrating structured geospatial knowledge into LLMs, thereby extending their reasoning capabilities to spatially informed domains.

## 1 INTRODUCTION

The rapid advancement of large language models (LLMs) has significantly influenced various domains, including natural language processing, code generation, and multimodal reasoning. Nevertheless, the integration of LLMs into geospatial research remains underexplored, presenting substantial challenges and unmet user requirements. When endowed with strong geospatial understanding, LLMs can perform tasks such as generating precise locations—not solely based on learned textual descriptions, but by utilizing high-dimensional geospatial data. Furthermore, geospatial information can function as a novel modality, aligning with other modalities such as images or videos, thereby enabling models to perform localization from visual inputs and fully incorporate geographical and spatial information into their reasoning processes.

Achieving these goals requires not only architectural innovations but also the development of specialized training paradigms and benchmark datasets tailored for geospatial applications. To this end, we propose a novel method to inject geospatial knowledge into language models, along with a systematic workflow comprising geospatial encoding, pretraining, fine-tuning, and post-training stages, to develop a domain-specialized LLM with embedded geospatial intelligence. Our approach builds upon the Qwen2.5-7B-Instruct architecture, enhanced through targeted adaptation to geospatial data modalities and knowledge sources. After training, our model successfully recognizes the introduced special tokens and their associated geospatial information, enabling effective application to real-world problem-solving.

## 2 RELATED WORK

In recent years, large language models (LLMs) have garnered significant attention due to their strong performance in natural language processing and wide applicability across domains such as healthcare, finance, and law. However, research on LLMs in the geographic domain remains limited and is still in its early stages.

Early studies Roberts et al. (2023); Mai et al. (2023) evaluated the capabilities of advanced closed-source models—such as ChatGPT and GPT-4—in geospatial tasks, primarily focusing on performance assessment and identifying key limitations in complex, context-sensitive geographic reasoning. GeoGPT enhances LLMs by integrating retrieval-augmented generation (RAG) and geospatial

libraries to support GIS workflows and satellite image analysis (Zhang et al., 2023). Similarly, GeoLLM employs fine-tuning techniques to extract geospatial knowledge from pre-trained LLMs for solving specific geospatial problems (Manvi et al., 2023). (Ding et al., 2023) try to do the POI-query matching task with LLM pretraining. Recent work (Liang et al., 2025; Zhang et al., 2025; Hsu et al., 2024) has explored multimodal approaches that leverage remote sensing imagery, maps, and textual data through vision-language large models to improve geospatial understanding.

Nevertheless, most existing studies focus on macro-scale or statistical applications—such as population density estimation, environmental and disaster monitoring, and urban traffic analysis.We aim to incorporate geospatial modality information as a novel modality representation, which, upon alignment with other modalities, enables the emergence of a large-scale model inherently capable of sophisticated geospatial reasoning and expression.

## 3 METHOD

This section outlines the key techniques and methodologies employed in this work. Section 3.1 presents the encoding algorithm we designed in this study. Section 3.2 elaborates on the continuous pre-training procedure aimed at equipping the model with general geospatial awareness, including data construction strategies and principles of knowledge integration; the resulting model, GSLLM-Base, is obtained through training in this phase. Section 3.3 presents a real-world application of our geospatial model. We designed a tailored pipeline comprising continuous pretraining, supervised fine-tuning, and reinforcement learning to enhance the model's performance on the geocoding task. The overall workflow is illustrated in Figure 4.

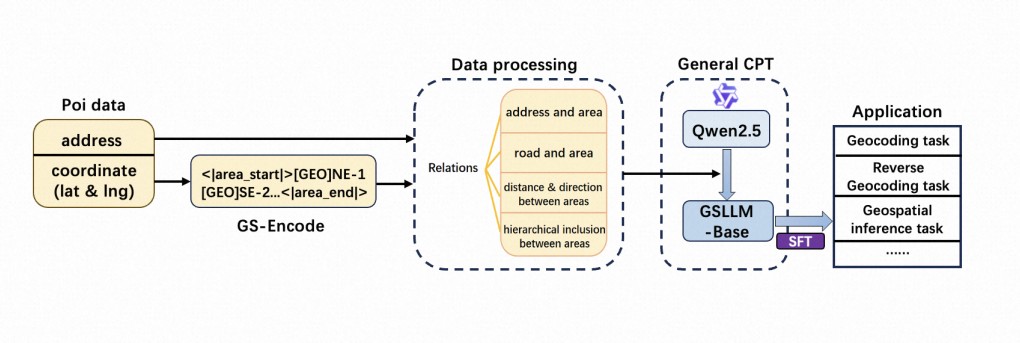

Figure 1: Overall workflow

### 3.1 GS-ENCODING

For LLMs, the original tokenization mechanism is ill-suited for effectively representing geographic coordinate information, specifically latitude and longitude. Hence, it is essential to design a more appropriate encoding method to enhance the model's understanding of spatial data. In this domain, Geohash is the most widely adopted algorithm. Building upon this foundation, we propose an improved encoding algorithm with enhanced expressive power that explicitly incorporates directional information. The overall encoding process is illustrated in Figure 1. The procedure proceeds as follows: First, starting from the entire Earth, we define west longitude and south latitude as negative, resulting in an initial longitude interval of [-180,180] and latitude interval of [-90,90]. This initial interval is then recursively subdivided into four equal quadrants–northeast (NE), southeast (SE), northwest (NW), and southwest (SW)—by bisecting both horizontally and vertically. The current bit of the encoding is determined according to which quadrant contains the target coordinate. Subsequently, the selected quadrant becomes the new interval and is further partitioned in the same manner to compute the next bit. This recursive partitioning continues until the desired encoding length is reached. For clarity, each bit in the encoding is formally defined as:

$$[GEO]\{direction\}\text{-}\{i\} \quad \{direction\} \in \{NE, SE, NW, SW\}, \{i\} \in \{1, 2, ...n\}$$

where n denotes the target encoding length, set to a maximum of 20 in our work, yielding 80 distinct token values across all positions and directions. These tokens are incorporated into the model's tokenizer to expand its vocabulary. After generating the sequence of encoding bits, we delimit the entire sequence using two special tokens, `<|area_start|>` and `<|area_end|>`, to form a structured representation of the geographic area. Figure 2 illustrates the complete encoding process.

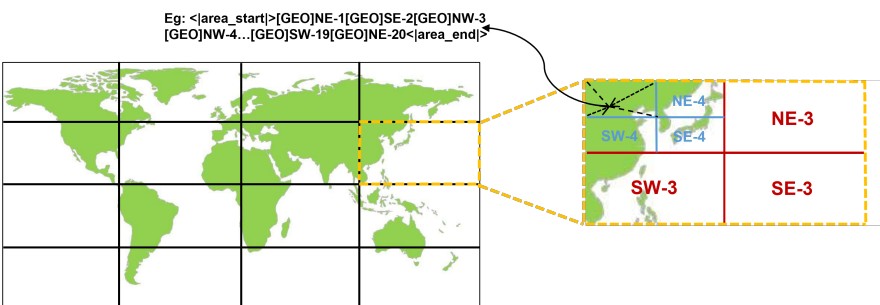

Figure 2: our encoding algorithm

According to the aforementioned principles, an increase in encoding bits leads to higher spatial accuracy. For instance, a 20-bit encoding corresponds to a rectangular area of approximately 20 meters × 40 meters, which meets the accuracy requirements of real-world applications. Moreover, since areas sharing a common prefix are hierarchically contained within larger areas, our experiments primarily adopt 20-bit encoding for high-precision positioning, while shorter encodings are utilized to represent broader areas, thereby preserving hierarchical spatial relations.

Our encoding algorithm overcomes the limitations of the GeoHash algorithm in preserving local proximity. As illustrated in Figure 3, two small areas (blue and yellow) are deemed distant under the GeoHash scheme due to a discrepancy in a high-order bit. However, in reality, these areas progressively converge through subsequent subdivisions and ultimately become adjacent, with a minimal actual spatial distance between them. By incorporating directional information at each bit level, our encoding enables the model—after training—to recognize the gradual spatial convergence of such areas, thereby effectively mitigating the boundary distance error inherent in GeoHash.

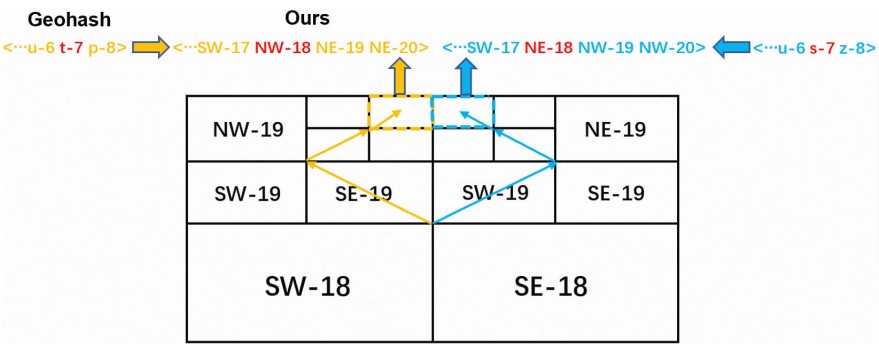

Figure 3: boundary distance error issue

## 3.2 GENERNAL CONTINUOUS PRE-TRAINING

In this stage, we aim to inject diverse geographic knowledge into a large language model (LLM) via continuous pre-training, thereby endowing it with robust geospatial perception capabilities. Specifically, the model should be able to recognize containment relations between addresses and areas, spatial distributions and positional relationships between areas and roads, as well as directional, distance, and containment relations among different areas. Building upon the POI data and encoding algorithm introduced in Section 3.1, we design multiple types of geospatially informative training

samples to facilitate the acquisition of these capabilities. Our data is primarily sourced from the company's internal POI database. A POI (Point of Interest) constitutes a fundamental geographic entity in online mapping services, representing specific physical locations and including attributes such as name, address, coordinates, and entity type. The detailed data construction methodology is outlined as follows:

### 3.2.1 ADDRESS-AREA RELATION

Injecting knowledge about the relationship between addresses and areas into large language models (LLMs) is crucial for enabling effective localization tasks. For each POI data entry, we first extract its address information and then apply our encoding algorithm to determine the corresponding area. To reinforce this relationship, we employ a bidirectional approach. Based on this framework, we construct a comprehensive set of templates to enhance linguistic diversity through matching. The specific template designs are detailed in Section A.1.

### 3.2.2 ROAD-AREA RELATION

For maps, roads or streets serve as their "meridians and collaterals," linking individual addresses. Thus, incorporating knowledge about the spatial distribution and positional relationships between roads and areas into large language models (LLMs) is crucial. For each area, we employ two methods to identify nearby roads: first, we utilize Qwen2.5-32B-Instruct(Team, 2024) to extract potential road names directly from the addresses within the area; second, we retrieve roads located within a 200-meter radius of the area's geographic coordinates. The results from both approaches are then merged and deduplicated. Similarly, we enhance the relational representation through dual-directional integration. The detailed construction methodology is provided in Section A.1.

### 3.2.3 DISTANCE AND DIRECTION RELATION BETWEEN AREAS

We also need to inject knowledge regarding the directional and distance relationships between areas into the LLM to enhance its perception of the overall geographic space. The data format is as follows: for any two areas, one can reach the other by moving a specific distance in a given direction. Through extensive learning of such relational knowledge, the model can effectively address the boundary distance error issue described in Section 3.1.

### 3.2.4 HIERARCHICAL INCLUSION RELATION BETWEEN AREAS

We further require injecting knowledge regarding the size of areas and their hierarchical inclusion relationships into the LLM. In addition to employing 20-bit encoding for fine-grained address localization, our model must also be capable of identifying larger areas, such as major commercial pedestrian streets. Specifically, we collect co-located addresses within a broader area and extract the common prefix from their 20-bit encodings to derive a shorter representation for the encompassing area. These hierarchical relations are then described in natural language to construct the pre-training dataset.

## 3.3 APPLICATION: GEOCODING

In order to evaluate the effectiveness of our pretrained model, we fine-tuned it on a representative downstream task based on GSLLM-base. The geocoding task is one of the most fundamental and critical tasks in the geographic domain, aiming to convert textual addresses into geographic coordinates for spatial localization. (Zandbergen, 2008)

### 3.3.1 DOMAIN-ADAPTIVE PRETRAINING

In our framework, the model localizes an address to an area represented by a 20-bit encoding, which is then converted back into coordinates to yield the final location. However, due to the data distribution and hierarchical inclusion relationships among areas, the localization error rate tends to increase with the number of encoding bits, as higher bit counts correspond to finer spatial subdivisions. More significantly, errors at higher-order bits result in greater positional inaccuracies. To mitigate this issue, we propose a method that enables the model to focus on error-prone bit positions

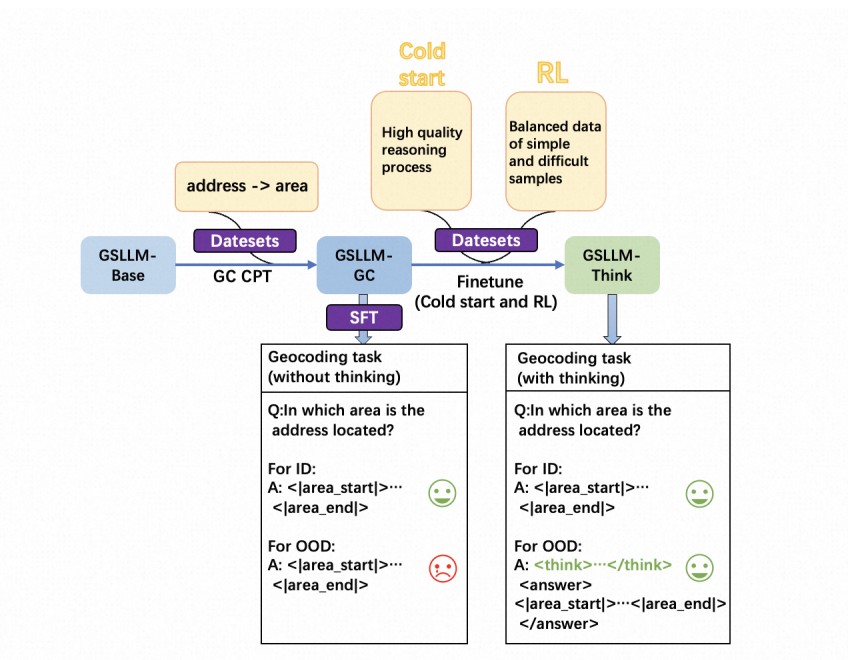

Figure 4: Overall workflow

during training by prioritizing their optimization. Specifically, we introduce weighted positions in the autoregressive loss calculation. Figure 5 illustrates the workflow of the weighting algorithm, and Section A.2 provides detailed weight specifications.

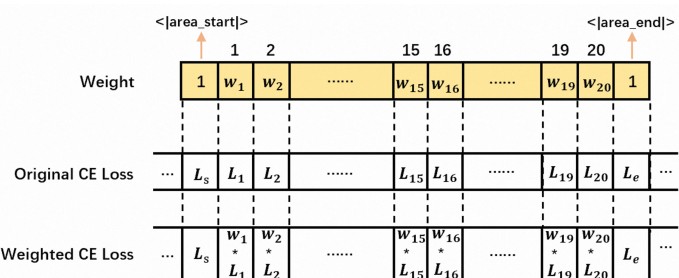

Figure 5: the weight algorithm

### 3.3.2 SUPERVISED FINETUNING(SFT)

In this phase, we construct high-quality and well-structured reasoning process data to facilitate the model's acquisition of the reasoning format. Our objective is to emulate the cognitive process employed by humans when determining a location. Through analysis of prevalent address formats in the dataset, we observe that the majority of addresses adhere to specific patterns. Statistical analysis reveals that over 80% of addresses can be classified into two distinct formats:

(1) A specific number on the road (e.g., No. 5 Chaoyang North Road).

(2) A certain distance in a particular direction from the reference position (e.g., 50 meters east of the intersection of Chaoyang North Road and Chaoyang West Road).

We can construct the reasoning process for these patterned addresses in a reasonable and efficient manner. Specifically, we first identify a reference address associated with the destination address and determine its corresponding area. Subsequently, by leveraging the distance and directional re-

lationship between the destination address and the reference address, we infer the area in which the destination address is located. A concrete construction example is illustrated in Figure 6. Furthermore, the model exhibits a certain level of accuracy when generating direct answers without explicit reasoning. To preserve this capability during reinforcement learning, we include a portion of non-reasoning samples in the cold-start dataset. This strategy enables the model to maintain a balance between reasoning-based and direct response generation after the cold-start phase.

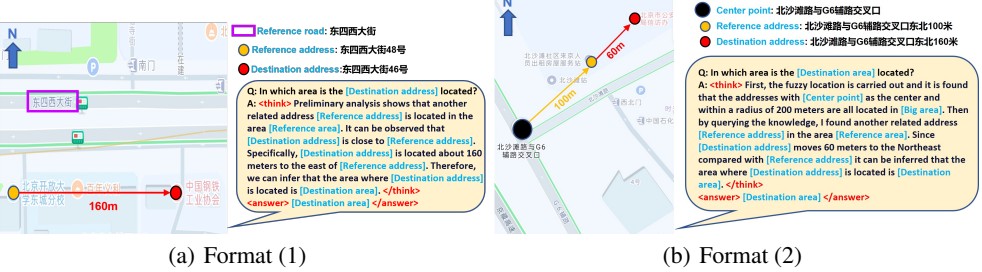

(a) Format (1)    (b) Format (2)

Figure 6: Example of reasoning process

### 3.3.3 REINFORCEMENT LEARNING

Following the cold start phase(SFT), we leverage reinforcement learning with the GRPO algorithm (Shao et al., 2024) to further activate the reasoning potential of the models. Specifically, for a question $q$ from the training dataset $D$, GRPO samples a group of responses $O = \{o_i\}_{i=1}^{G}$ from the old policy $\pi_{old}$ and then optimizes the policy model by maximizing the following objective:

$$\mathcal{J}(\theta) = \mathbb{E}_{(q)\sim D, \{o_i\}_{i=1}^{G} \sim \pi_{\theta_{old}}(O|q)}$$

$$\frac{1}{G}\sum_{i=1}^{G}\frac{1}{|o_i|}\sum_{t=1}^{|o_i|}\left\{\min\left[\gamma_{i,t}(\theta)\hat{A}_{i,t}, \text{clip}\left(\gamma_{i,t}(\theta), 1-\epsilon, 1+\epsilon\right)\hat{A}_{i,t}\right] - \beta\mathrm{D}_{KL}\left[\pi_\theta\|\pi_{ref}\right]\right\},$$

where $\gamma_{i,t}(\theta) = \frac{\pi_\theta(o_{i,t}|q,o_{i,<t})}{\pi_{\theta_{old}}(o_{i,t}|q,o_{i,<t})}$, $\pi_{ref}$ represents the reference model, and the term $D_{KL}$ introduces a KL divergence constraint to limit how much the model can deviate from this reference. The advantage estimate $\hat{A}_i$ measures how much better the response $o_i$ is compared to the average response, which is computed using a group of rewards $\{r_1, r_2, \ldots, r_G\}$ for the responses in set $O$: $\hat{A}_i = \frac{r_i - \text{mean}(\{r_1, r_2, \ldots, r_G\})}{\text{std}(\{r_1, r_2, \ldots, r_G\})}$.

The most crucial part of training GRPO lies in the calculation method of rewards. Unlike common mathematical reasoning problems, we first introduce a reward model to better evaluate the quality of the reasoning process, which is trained using artificially constructed pair-wise positive and negative samples. Specifically, the reasoning process for positive samples is constructed according to the method described in Section 3.3.2, while negative samples are generated by randomly altering key nodes in the reasoning process that are prone to errors. The output of the reward model is defined as $Reward_m$, which represents the score assigned to the quality of the reasoning process. Next, in designing the reward function $Reward_r$ for reasoning outcomes, we aim to satisfy the following criteria:

1. For simple samples where correct answers can be obtained without explicit reasoning, the model is encouraged to respond directly.

2. For complex samples where direct responses lead to incorrect answers, the model is incentivized to perform step-by-step reasoning to reach the correct solution.

Accordingly, the input to the result reward function comprises two components: prediction error distance and response length. Based on these considerations, we formulate the following function:

$$Reward_r = \begin{cases} R_d(d) + max(R_l(l)) + \beta & 0 \leq d \leq 200, l < 100 \\ R_d(d) + max(R_l(l)) & 0 \leq d \leq 200, l \geq 100 \\ R_d(d) + R_l(l) & 200 < d \leq 1000, 100 < l < 400 \\ 0 & d > 1000 \end{cases}$$

where d indicates prediction error distance, l indicates response length, $R_d(d)$ represents error reward function and $R_l(l)$ represents the length reward function. The specific formula is as follows:

$$R_d(d) = \begin{cases} -e^{(0.000016d^2)} + 2 & 0 \leq d \leq 200 \\ -\frac{1}{8000}d + \frac{1}{8} & 200 < d \leq 1000 \end{cases}$$

$$R_l(l) = -(\frac{1}{112500})l^2 + \frac{1}{255}l - 0.2555 \quad 100 \leq l \leq 400$$

Some specific values in the above formula are designed based on practical requirements. For instance, $d < 200$ indicates a correct answer, while $l < 100$ represents a direct response without reasoning. The peak of $R_l(l)$ is set at (250,0.3) because a response length around 250 typically corresponds to a well-structured reasoning process. Additionally, $\beta$ is a hyperparameter introduced to encourage the model to prefer direct response when the question can be answered correctly without reasoning. Experimental results confirm that setting $\beta = 2$ achieves a good balance between direct response and reasoning.

The total reward function is as follows, where the hyperparameter $\lambda$ is used to control the ratio between the output of result reward function and the reward model, we set it to 0.01:

$$Reward = Reward_r + \lambda * Reward_m$$

Regarding the data used during training, to ensure the model acquires both direct answering capability and reasoning-based responding, the dataset must be carefully balanced in composition. As previously discussed, the goal is for the model to answer directly on simple samples while engaging in reasoning for more challenging ones. However, if the dataset predominantly consists of simple samples, the model tends to favor direct responses during exploration; experimental results indicate that the average response length rapidly decreases, eventually converging almost entirely to direct answers. Conversely, an overabundance of difficult samples may lead the model to over-rely on reasoning pathways. To address this imbalance, we employ GSLLM-GC prior to cold-start to generate responses for candidate data intended for reinforcement learning, labeling incorrect outputs as difficult samples and correct ones as simple samples. We then construct the reinforcement learning dataset by combining simple and difficult samples in approximately a 1:1 ratio. The overall construction process is illustrated in Figure 7

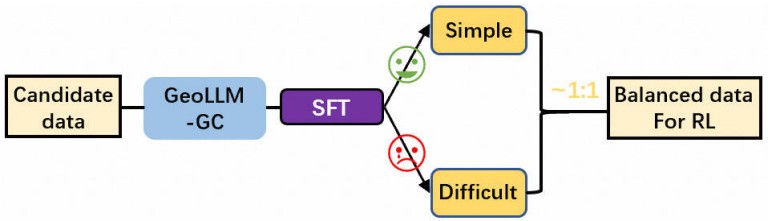

Figure 7: Construction method of reinforcement learning training set

# 4 EXPERIMENTS AND EVALUATION

This section introduces the overall experimental design of our work, as well as the methods and results for evaluating the model's capabilities. Section 4.1 introduces the collection and construction process of our dataset. Section 4.2 describes the specific settings for each experiment. Section 4.3 evaluates the model's performance on the Geocoding task. The evaluation is divided into two parts: In-Distribution (ID) testing on addresses seen during continuous pre-training to measure the model's memorization ability, and Out-of-Distribution (OOD) testing on addresses unseen during continuous pre-training to assess its generalization capability.

## 4.1 DATASET

We select all POI data from Beijing, China, for our experiments. After applied a series of data cleansing and filtering strategies, we got approximately 2.8 million addresses. Using our encoding algorithm, these addresses are assigned to around 1 million areas with 20-bit encoding. From them, we extract some addresses for OOD testing and for reinforcement learning training. Based on the data construction method described in Section 3.2, we created a dataset named dateset-CPT, containing approximately 100 million tokens, for the general continuous pre-training. For the specialized Geocoding continuous pre-training introduced in Section 3.3, we generated one data entry for each address used in dateset-CPT according to the required format, thereby constructing a Geocoding description dataset named dataset-GC for this phase of pre-training. For Section 3.3.2, we first construct about 1w high-quality reasoning process datas for cold-start. Then, approximately 2w pairs of pair-wise data are constructed using the method described in Section 3.3.3 to train the reward model. Finally, the balanced dataset with a total of 10w is constructed using the method shown in Figure 7 for reinforcement learning.

## 4.2 EXPERIMENTS SETUP

We choose Qwen2.5-7B-Instruct(Team, 2024) as the starting base model. For Section 3.2, we perform 5 epochs of pre-training on the dateset-CPT dataset based on Qwen2.5-7B-Instruct with the learning rate set to 1e-4, resulting in GSLLM-Base. For Section 3.3.1, we perform 2 epochs of pre-training on the dataset-GC dataset based on GSLLM-Base with the learning rate set to 1e-4, resulting in GSLLM-GC. For Section 3.3.2, we perform 2 epochs of Supervised Fine-Tuning (SFT)(Radford et al., 2018) on GSLLM-GC using high-quality reasoning process datas with the learning rate set to 5e-6 to complete cold-start. For Section 3.3.3, We first perform 3 epochs of LoRA(Vaswani et al., 2017) fine-tuning on the cold-started model using pair-wise data with the learning rate set to 1e-5 to obtain the reward model. Then, we conduct 1 epoch of reinforcement learning training on the constructed balanced dataset, ultimately yielding GSLLM-Think.

## 4.3 GEOCODING TASK EVALUATION

### 4.3.1 GC-INDEXING

There is extensive research in this field, most of which relies on various types of POI retrieval methods, such as (Huang et al., 2024). Our baseline method employs an online geocoding (GC) service that uses a search engine to match the input address with entries in the database and retrieve the corresponding coordinates. We did not adopt general-purpose large language models such as qwen-max because they are fundamentally incapable of accurately retrieving or resolving precise address information.

### 4.3.2 RESULTS & ANALYSIS

| Task | Source | GC-Indexing | GSLLM-Base | GSLLM-GC | GSLLM-Think |
|------|--------|-------------|------------|----------|-------------|
| Geocoding | ID | 0.89 | 0.83 | 0.94 | 0.92 |
| | OOD | 0.71 | 0.71 | 0.79 | **0.81** |

Table 1: result of Geocoding

Table 1 presents the test results of the geocoding task. We adopted a prediction error distance of less than 200 meters as the criterion for correctness and compared the performance of baseline methods with our model. Compared to the retrieval-based baseline, our best model achieved approximately a 20% improvement in accuracy. GSLLM-Think demonstrates the capability to reason through complex problems while retaining the ability to provide direct responses to simpler queries, effectively balancing the thinking and straight-forward answering mode.

For the pretrain phase, we hypothesized that data density could significantly affect the accuracy of the GC task. To evaluate this hypothesis, we trained our model on multiple datasets with varying sizes. The results are summarized in Figure 8 and Figure 9. The study area—Chaoyang

| Region | address num | Accuracy |
|---|---|---|
| | 80k | 0.32 |
| | 200k | 0.56 |
| Chaoyang District | 450k | 0.65 |
| | 680k | 0.73 |
| | 750k | 0.73 |

Figure 8: Accuracy vs data size.

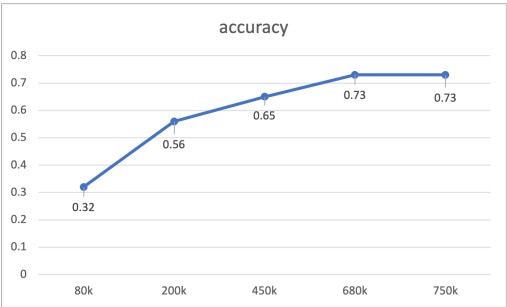

Figure 9: Performance trend.

District($\sim 455$ km$^2$), Beijing—contains over one million POIs. Training began with 80,000 addresses, yielding an accuracy of 32%. As the training sample size increased to 680,000, accuracy plateaued at approximately 73%; further increases in data volume did not lead to significant improvements in OOD inference performance. Consequently, we sampled the training set at this ratio($\sim 70\%$) to balance computational cost and accuracy.

## 5 ABLATION STUDIES

### 5.1 EFFECTIVENESS OF WEIGHTING METHOD IN GEOCODING CONTINUOUS PRE-TRAINING

In Section 3.3, during the continuous pre-training focused on geocoding using the dataset-GC dataset, we design a weighting method to achieve finer and more effective optimization. To validate the effectiveness of this method, we conduct a comparative experiment using the same dataset-GC dataset for continuous pre-training without the weighting method, while keeping all hyperparameters identical. Table 2 presents the experimental comparison results. The results show that, compared to the non-weighting method, the use of the weighting method leads to significant improvements on both the ID and OOD test sets. This demonstrates that our weighting method is highly effective in achieving more refined geocoding optimization.

| Task | Source | GSLLM-GC (w weighting) | GSLLM-GC (w/o weighting) |
|---|---|---|---|
| Geocoding | ID | 0.94 | 0.88 |
| | OOD | 0.79 | 0.76 |

Table 2: Effectiveness of weighting method

## 6 CONCLUSION

In this work, we propose a novel method that leverages the separation strategy from GeoHash and a new encoding system to inject geospatial knowledge into large language models (LLMs). The model is pretrained on point-of-interest (POI) data—comprising names, addresses, and coordinates—to obtain GSLLM-base. Subsequently, the model is adapted to several real-world scenarios through supervised fine-tuning (SFT) and reinforcement learning (RL) based on GSLLM-base. Our results demonstrate that the proposed approach significantly mitigates the challenge of LLMs learning location information from floating-point coordinates.

The framework supports the integration of additional data modalities, such as images, into the workflow. Several applications are being developed based on this model, including reverse geocoding—i.e., generating textual addresses from geographic coordinates—and other location-aware tasks. In this study, we have demonstrated the feasibility of GSLLM; future work will extend its coverage to broader regions through larger models and more extensive datasets.

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

# A APPENDIX

## A.1 DATA FORMAT

This section introduces some specific details of data construction in the main text. First, in Section 3.2.1, we designed over 50 matching templates to construct the relation between address and area, with several examples illustrated in Figure 10. Then fill in the specific "area" and "addresses" information according to the template.

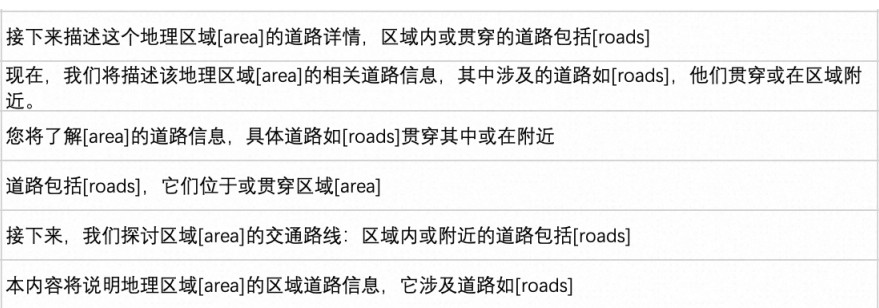

| 现在描述这个区域[area]的地址详情，其中包含的地址包括[addresses] |
| 以下是该位置[area]的地址信息，区域内地址有[addresses] |
| 在地理区域[area]中，包含地址[addresses]这几个地址 |
| 地址列表:[addresses]，这些地址都包含在[area]标识的区域中 |
| 在区域[area]中，我们可以找到以下地址[addresses] |
| 在地理区域[area]中，您可以找到以下地址[addresses] |
| 我们将区域[area]的地址进行了汇总，结果如下：[addresses] |
| 地址如[addresses]均属于区域[area] |
| 现在我们来看一下[area]的地址概况，涉及的主要有:[addresses] |

Figure 10: data templates for Section 3.2.1

For Section 3.2.2, we designed over 20 matching templates to construct the relation between road and area, with several examples illustrated in Figure 11. Then fill in the specific content as well. Furthermore, to preserve the model's natural language ability, we incorporate a portion of data

| 接下来描述这个地理区域[area]的道路详情，区域内或贯穿的道路包括[roads] |
| 现在，我们将描述该地理区域[area]的相关道路信息，其中涉及的道路如[roads]，他们贯穿或在区域附近。 |
| 您将了解[area]的道路信息，具体道路如[roads]贯穿其中或在附近 |
| 道路包括[roads]，它们位于或贯穿区域[area] |
| 接下来，我们探讨区域[area]的交通路线：区域内或附近的道路包括[roads] |
| 本内容将说明地理区域[area]的区域道路信息，它涉及道路如[roads] |

Figure 11: data templates for Section 3.2.2

generated by advanced LLM like Qwen-Max during the data construction process. This inclusion enables the model to produce more diverse and rich content, mitigating the risk of over-reliance on template-matching data that could lead to an excessive focus on fixed formats. **??** illustrates examples of the prompts used in constructing the data.

## A.2 WEIGHT DETAIL

Based on practical requirements, in the weighting algorithm, we assign weights to the 20-bit encoding of the area in descending order of importance, as detailed below: {2,2,2,2,2,2,2,2,128,128,64,64,32,32,16,16,8,8,4,4}

