# OpenReview forum: "GSLLM: Geospatial Knowledge Acquisition for Large Language Model"
_ICLR.cc/2026/Conference — Submitted to ICLR 2026_

### Official Review · Reviewer_r61t · 2025-10-27

**Soundness:** 1
**Presentation:** 2
**Contribution:** 1
**Rating:** 2
**Confidence:** 4

**Summary:**

The paper presents GSLLM, a framework for integrating geospatial knowledge into LLMs. It introduces a geospatial encoding algorithm that extends GeoHash by embedding directional information into discrete tokens. The framework follows a three-stage process, continuous pre-training, SFT, and RL, to equip the model with spatial reasoning abilities. GSLLM is trained on large-scale POI data from one city, Beijing. Experimental results show performance gains on geocoding tasks compared to intra-model comparison, and ablation results show the benefit of a proposed loss-weighting scheme.

**Strengths:**

The paper introduces a direction-aware hierarchical encoding extending GeoHash and claims improved spatial locality.

The model combines continuous pre-training, SFT, and reinforcement learning, which is a multi-stage training pipeline.

The paper defines multiple relations, such as address-area, road-area, distance-direction, and hierarchical inclusion, to construct structured geospatial knowledge.

**Weaknesses:**

The proposed direction-based encoding resembles common discrete grid partitioning schemes (e.g., Quadtree Grid), and its novelty is limited. Moreover, if the base LLM lacks inherent geographic understanding as the paper claims, then how the model knows that regions such as NW-18 and NE-18 in Fig. 3 are spatially adjacent?

Many critical choices, such as the 20-bit encoding length, positional loss weights, and the 200 m threshold in the reward function, are fixed heuristically without quantitative analysis or sensitivity experiments.

No comparison with existing geospatial LLMs (GeoGPT, GeoLLM, etc.) despite referencing them in Sec. 2.

Claims of “enhanced geospatial reasoning” are not substantiated by multi-task or cross-domain results. The experiments focus solely on coordinate retrieval within a single city, which does not align with the paper’s claim of broad downstream applicability or demonstrate generalized geospatial reasoning ability.

The ablation study only evaluates the effect of loss weighting, while other key components, such as the proposed geospatial encoding method and the various relations, lack corresponding ablations, leaving their individual contributions unclear.

**Questions:**

If the proposed direction-based encoding were replaced with a GeoHash representation, how’s the model’s performance

---

### Official Review · Reviewer_K6ds · 2025-11-01

**Soundness:** 2
**Presentation:** 2
**Contribution:** 3
**Rating:** 4
**Confidence:** 3

**Summary:**

This paper proposes the GSLLM framework to inject geospatial knowledge into large language models. By designing specialized geospatial encoding tokens to represent coordinates, locations, and addresses, the framework embeds geospatial information into the model's token space. Built upon Qwen2.5-7B-Instruct, the model demonstrates strong spatial reasoning capabilities on downstream tasks such as geocoding through continuous pre-training, supervised fine-tuning, and reinforcement learning.

**Strengths:**

1. Enabling large language models to comprehensively understand geospatial knowledge is a highly important research problem with significant practical implications.

2. The proposed GS-encoding method is easy to understand and provides valuable insights for domain researchers.

**Weaknesses:**

1. Lack of baseline comparisons: The paper does not compare against general-purpose LLMs. What are the performance levels of models such as GPT-5, Claude Sonnet 4.5, or Qwen-Plus on the same dataset? How do smaller-scale LLMs (7B-32B parameters) perform on these tasks?

2. Missing analysis on general NLP capabilities: After training and fine-tuning, does the model's performance on general natural language processing tasks degrade? This potential trade-off is not evaluated.

3. The overall presentation focuses heavily on describing the workflow. Providing more discussion on design motivations, failure cases, and ablation studies would be highly beneficial.

**Questions:**

If the authors can address the three weaknesses mentioned above, I would be willing to revise my score accordingly.

---

### Official Review · Reviewer_y1r4 · 2025-11-03

**Soundness:** 2
**Presentation:** 3
**Contribution:** 2
**Rating:** 6
**Confidence:** 2

**Summary:**

Introduces GSLLM, a framework for integrating geospatial knowledge into large language models. The key idea is a novel encoding scheme (GS-Encoding) that represents coordinates as hierarchical directional tokens (e.g., [GEO]NE-1, [GEO]SW-2, …), enabling spatial reasoning within the model’s token space. Building on Qwen2.5-7B-Instruct, continuous geospatial pretraining is performed, followed by supervised fine-tuning and reinforcement learning (GRPO) for a geocoding task (address → coordinates). Experiments on a 2.8 M-entry Beijing POI dataset show substantial accuracy gains over retrieval-based baselines (20% absolute improvement), and ablations demonstrate the benefits of positional weighting and hierarchical encoding.

**Strengths:**

1. Novel representation: The GS-Encoding method extends GeoHash with explicit directional information, mitigating boundary-distance errors and preserving spatial hierarchy.
2. Practical relevance: Demonstrates measurable improvement on a key real-world task (geocoding), with potential for downstream spatial reasoning and reverse-geocoding.
3. Targeted pipeline for a specific application: Clear multi-stage training (pretrain then SFT and finally RL) that combines structured geospatial data with language understanding.

**Weaknesses:**

1. Limited geographic scope: All experiments use Beijing POI data; cross-region generalization remains untested.
2. Baseline simplicity: The comparison is only to a retrieval-based GC-Indexing method—omitting strong learned baselines (e.g., GeoLLM, GeoGPT).
3. Clarity issues: Some figures and variable names (e.g., weighting scheme, reward terms) could be better explained or unified with the text.
4. Training details: The GRPO reward function appears hand-tuned; sensitivity analysis or justification of constants (β, λ) would strengthen confidence.

**Questions:**

1. How does GS-Encoding generalize to unseen regions or different coordinate systems (e.g., WGS84 vs GCJ02)?
2. Could the model’s learned geospatial embeddings be evaluated on auxiliary spatial reasoning benchmarks (e.g., distance estimation, containment)?
3. Are there privacy or licensing considerations with the internal POI data? This may be relevant to reproducibility.

---

### Official Review · Reviewer_KoPT · 2025-11-05

**Soundness:** 1
**Presentation:** 2
**Contribution:** 2
**Rating:** 2
**Confidence:** 4

**Summary:**

This paper introduces GSLLM, which is designed to imbue LLMs with geospatial knowledge across the training pipeline (pre-training, supervised fine-tuning, RL pos-training). The core of the method is to introduce an encoding scheme which represents geographic locations (i.e. latitude, longitude) via special tokens.

**Strengths:**

**(S1)**: The GS-Encoding scheme is interesting. It makes sense to expand the vocabulary to encode geospatial information more effectively.

**(S2)**: An ablation study on the weighted loss is performed.

**Weaknesses:**

Overall, this paper feels incomplete. The method feels over-engineered, and it's not clear to me what the benefits are of this approach as almost no baselines from the literature were used. After reading this paper, I was not able to grasp the motivation or the core problem that that this method tackles, or whether the proposed approach is effective. To go into more detail:

**(W1)**: Insufficient experimental validation. The only visible baseline was in Table 1, GS-Indexing. This is described as (L410) an "online geocoding (GC) service that uses a search engine". This is extremely vague, and doesn't offer any detail or citation. Any scientific experiment needs to provide detail on the baseline and experimental setup. This is a glaring omission. Another omission is a comparison against a Retrieval-Augmented Generation (RAG) baseline. Given that the related work mentions RAG-based approaches (like GeoGPT ), a comparison of GSLLM against the same base model (Qwen2.5-7B) augmented with a RAG pipeline retrieving from the same POI database is essential to justify the complexity of the proposed knowledge-injection method.

**(W2)**: Framework design. It is absolutely not clear to me why pre-training, SFT, and RL is required for this task. It seems excessive and no clear motivation was provided for this pipeline. Was an ablation performed to demonstrate the necessity of each stage? Why is RL with a trained reward model necessary? It seems to me that just SFT on high-quality data should be enough. The rest of the design feels over-engineered. In fact, GeoLLM clearly demonstrate that SFT on a few high-quality samples is enough to elicit the latent geospatial information in LLMs and enables them to perform quite effectively on many geospatial tasks. This is another missing baseline in the experiments. Moreover, a much simpler RAG approach might be equally or more effective. The paper does not fully justify why this deep "baking-in" of knowledge is superior to on-demand retrieval

**(W3)**: Potentially biased RL data curation: The method for balancing the RL dataset (Figure 7) involves using the intermediate model (GSLLM-GC) to label candidate data as "simple" (correct answers) or "difficult" (incorrect answers). This strategy seems circular and may introduce bias. A sample is labeled "difficult" not because it is inherently complex, but simply because the intermediate model failed on it.

**(W4)**: Limited geographic and linguistic scope. The experiments are exclusively based on POI data from Beijing, China. This represents a single language (Chinese) and a single, relatively consistent address format. The GS-Encode system, however, is designed to be global. It is unclear how this framework would scale to other languages and a wide variety of global address formats.


Overall, this paper does not meet the bar for an ICLR publication.

**Questions:**

**(Q1)**: L163 "Our data is primarily sourced from the company’s internal POI database."  -- What does this mean? Some description or examples of the dataset would be quite relevant.

**(Q2)**: L172 "To reinforce this relationship, we employ a bidirectional approach" What does this mean?

**(Q3)**: L203 -- Can you provide a concrete example of the geocoding task? What are the inputs and outputs?

**(Q4)**: L264 -- Can you provide a citation for this claim?

**(Q5)**: L388 -- what is 1w, 2w??

---

### Meta-Review · Area_Chair_aHFi · 2026-01-06

**Summary:**

This paper proposes GSLLM, a framework for injecting geospatial knowledge into LLMs via specialized tokens and multi-stage training. The core innovation is GS-Encoding, a direction-aware hierarchical encoding scheme extending GeoHash to represent coordinates/locations as discrete tokens, mitigating boundary-distance errors. Built on Qwen2.5-7B-Instruct, the pipeline includes continuous pre-training (on POI data for address-area, road-area, and spatial relations), supervised fine-tuning (SFT) with reasoning process data, and reinforcement learning (RL via GRPO) to optimize geocoding performance. Experiments on 2.8M Beijing POI entries show a 20% accuracy improvement over a retrieval-based baseline, with ablations validating the weighted loss for geocoding pre-training.

Reviewers recognized the practical relevance of geospatial LLM enhancement and the novelty of GS-Encoding. Key concerns centered on insufficient baseline comparisons, limited experimental scope, over-engineered training pipeline, and lack of ablation for core components. Due to the absence of the rebuttal, all concerns remain and the paper is not ready to be accepted at the moment.

**Reviewer Concerns:**

Key Concerns Raised:
- Baseline inadequacy (all reviewers): Only compares to a vague "retrieval-based GC-Indexing" baseline; omits critical comparisons with state-of-the-art geospatial LLMs (GeoLLM, GeoGPT), RAG-augmented versions of the same base model, and general-purpose LLMs (GPT-5, Claude Sonnet 4.5).
- Experimental scope limitations (KoPT, y1r4, r61t): Exclusively uses Beijing POI data (single language, uniform address format); no cross-region, cross-language, or multi-task validation (e.g., distance estimation, containment reasoning) to substantiate "general geospatial reasoning" claims.
- Training pipeline justification (KoPT, r61t): No ablations to validate the necessity of RL or multi-stage pre-training/SFT/RL; RL data curation (labeling "difficult" samples via intermediate model) risks circular bias.
- Component and parameter clarity (r61t, y1r4): Lack of ablations for GS-Encoding (vs. standard GeoHash) and spatial relation data; heuristic parameter choices (20-bit encoding, 200m accuracy threshold, RL reward weights) without sensitivity analysis.
- Presentation and reproducibility (KoPT, K6ds): Vague dataset descriptions (internal POI database with no examples), unclear terminology (e.g., "bidirectional approach" for address-area relations), and missing details on privacy/licensing of proprietary data.

Addressed in Rebuttal:
- None (no author rebuttal provided in the materials).

Outstanding:
- All key concerns remain unaddressed due to the absence of a rebuttal. Critical gaps in baseline comparisons, experimental generalization, pipeline justification, and component ablation are unresolved.

**Reviewer Scores:**

KoPT: 2 → Remains 2

y1r4: 6 → Remains 6

K6ds: 4 → Remains 4

r61t: 2 → Remains 2

---

### Decision · Program_Chairs · 2026-01-26

Reject